# HMGN1 enhances CRISPR-directed dual-function A-to-G and C-to-G base editing

Chao Yang[1,2], Zhenzhen Ma[3], Keshan Wang[4], Xingxiao Dong[5], Meiyu Huang[6], Yaqiu Li[1,2], Xiagu Zhu[7], Ju Li [8], Zhihui Cheng[3], Changhao Bi [1,2] ✉ & Xueli Zhang [1,2] ✉

C-to-G base editors have been successfully constructed recently, but limited work has been done on concurrent C-to-G and A-to-G base editing. In addition, there is also limited data on how chromatin-associated factors affect the base editing. Here, we test a series of chromatin-associated factors, and chromosomal protein HMGN1 was found to enhance the efficiency of both C-to-G and A-to-G base editing. By fusing HMGN1, GBE and ABE to Cas9, we develop a CRISPR-based dual-function A-to-G and C-to-G base editor (GGBE) which is capable of converting simultaneous A and C to G conversion with substantial editing efficiency. Accordingly, the HMGN1 role shown in this work and the resulting GGBE tool further broaden the genome manipulation capacity of CRISPR-directed base editors.

CRISPR-directed base editors (BEs) including a deoxynucleotide deaminase and a catalytically impaired Cas9 or Cpf1 can introduce single-base conversion. The recently developed CBE[1], ABE[2], and GBE/CGBE[3–5] enable programmable C-to-T, A-to-G, and C-to-G base editing, respectively. These technologies could edit the genome sequence without inducing a DNA double-strand break or requiring donor DNA templates, with exciting prospects in the genetic therapy of mutational-associated diseases. It was reported that ABE could directly correct the pathogenic single-base mutation of nuclear lamin A in cultured fibroblasts from children with progeria[6]. And CBE was successfully applied to edit BCL11A enhancer, which prevented sickle cell phenotype and attenuated the imbalanced globin chain in erythrocytes[7]. To broaden the application of this technology, we and other labs have been working on increasing the efficiency and diversifying the editing patterns of BEs. Richter et al.[8] greatly improved the editing efficiency of ABE through phage-assisted evolution. Koblan as well as our lab constructed and further enhanced the editing efficiency of C-to-G transition with DNA repair factors[9] or DNA binding proteins[10]. However, the genome editing capacity still has room for improvement.

One base editor normally catalyzes the transition for a single type of base pair, and it is not possible to simultaneously implement multiple editors in most scenarios. However, studies reported the discovery of MNVs (multi-nucleotide variants), which were recognized as two or more nearby variants existing on the same haplotype in an individual. Significantly, the MNVs were identified as a clinically and biologically important class of genetic variation, influencing the functional interpretation of genomic data[11,12]. To extend the editing possibilities and study MNVs-associated genetic diseases, researchers have developed dual-function base editors, which enable simultaneous C-to-T and A-to-G conversion in mammalian cells[13–16]. In addition, an AGBE system was constructed using human APOBEC3A and TadA, which could catalyze four types of base conversions[17]. These reconstructed BEs broadened the capability of base editing for applications in genetic therapy or gene regulation. However, a series of MNVs with simultaneous C-to-G and A-to-G mutations remain poorly understood, and related genetic correction studies are hindered by the lack of genome editing tools. For instance, it was reported that the concurrent C-to-G and A-to-G mutation in GAA (alpha-glucosidase) resulted in glycogen storage disease type II[18]. Additionally, the high efficiency and

[1]Tianjin Institute of Industrial Biotechnology, Chinese Academy of Sciences, Tianjin, China. [2]Key Laboratory of Systems Microbial Biotechnology, Chinese Academy of Sciences, Tianjin, China. [3]College of Life Sciences, Nankai University, Tianjin, China. [4]Department of Urology, Union Hospital, Tongji Medical College, Huazhong University of Science and Technology, Wuhan, China. [5]School of Biological Engineering, Dalian Polytechnic University, Dalian, China. [6]College of Life Sciences, Guangxi Normal University, Guilin, China. [7]College of Biotechnology, Tianjin University of Science and Technology, Tianjin, China. [8]College of Life Sciences, Tianjin Normal University, Tianjin, China. ✉e-mail: bi_ch@tib.cas.cn; zhang_xl@tib.cas.cn

specificity of base editors are indispensable for the construction of disease models and the development of genetic therapies for mutational disease. However, the recently developed AGBE showed low efficiency and specificity in performing concurrent C-to-G and A-to-G[17] conversions. Despite the prime editor that could introduce the concurrent C-to-G and A-to-G transition, normally the base editing could induce more efficient editing in a restricted editing range[4,19]. Furthermore, the operation of the base editor is more convenient than that of the prime editor. Thus, it is desirable further expand the diversity and capacity of BEs with a highly specific and efficient dual-function base editor, which simultaneously catalyzes C-to-G and A-to-G conversions.

It was reported that Cas9 nuclease activity was positively associated with elevated chromatin accessibility. Ding et al.[20] demonstrated that Cas9-dependent deletion or insertion was improved via fusion with chromatin-modulating peptides, particularly at refractory target sites. Furthermore, Liu et al.[21] found that transactivation modules could enhance Cas9-dependent editing efficiency, especially at nuclease-refractory target sites. Both of their functions are demonstrated to increase chromatin accessibility, which might promote Cas9 activity. More importantly, the editing efficiency of ABE and CBE was also enhanced with the inhibition of histone deacetylase to alter the chromatin state[22]. Collectively, these works indicate that genome editing is affected by the chromatin microenvironment. Chromatin remodelers and modifiers were reported to orchestrate the chromatin state, and thus influence the DNA repair process. It was demonstrated that BARD1 (BRCA1-associated RING domain protein 1) coordinates the binding of H2AK15ub and unmethylated H4K20 to promote homologous recombination[23], as well as corroborated the dimethylation of histone H3K36 and enhanced the repair of DNA double-strand breaks[24]. Moreover, LSD1 (lysine-specific histone demethylase 1)-mediated histone demethylation was reported to be involved in base excision repair induced by hydrogen peroxide[25]. Nevertheless, there were still only limited reports on how chromatin-associated factors influence base editing. Accordingly, we hypothesized that the fusion of chromatin-associated factors with BEs might increase the efficiency of base editing, which could be a promising strategy to further optimize BEs.

In this research, we engineer GBE and ABE by fusing them with a series of chromatin-associated factors, which optimizes the editing efficiency or indel frequency. Furthermore, we fuse HMGN1 with GBE, ABE, and GGBE for higher editing efficiency, lower indel byproducts or a unique editing pattern. Finally, we also demonstrated the application of GGBE in correcting or creating MNVs in mammalian cells.

## Results

### Analysis of chromatin-associated factors for improving GBE and ABE editing

To further improve the efficiency and expand the editing scope of base editors, we intended to optimize them by integrating chromatin-associated factors and further construct a specific dual-function C-to-G and A-to-G base editor (GGBE) (Fig. 1a). Accordingly, the nascent GBE[5] was selected for investigation of C-to-G editing, while the latest highly efficient A8e (ABE8e-V106W)[8] with reduced off-target editing was selected to explore A-to-G editing. Then we constructed a series of engineered GBE and A8e variants fused with 16 chromatin-associated factors, including chromatin-modulating factors, histone methylation factors, histone acetylation factors, and histone ubiquitination factors. Since previous work demonstrated the GBE variants with amino-terminal fusions exhibited the highest editing efficiency[26,27], this fusion pattern was also employed for the selected chromatin-associated factors (Fig. S1A). HEK293T cells were transiently co-transfected with the reconstructed GBE/A8e and gRNA vectors, and the editing efficiency within the editing window as well as the indel frequency were determined via high-throughput sequencing (HTS) and CRISPResso2[28].

Our data revealed that several chromatin-associated factors improved the C-to-G editing efficiency at the HIRA and EMX1 loci, especially the chromatin-modulating factors and histone ubiquitination factors. Notably, the HMGN1-fused GBE (HMGN1-GBE) showed the highest editing efficiency at position C6 of the protospacer, which was also superior to SadN-GBE, an enhanced GBE variant (Fig. 1b). The HMGN1-fused A8e (HMGN1-A8e) also exhibited the highest editing efficiency, but was similar or slightly higher to the control at the HEK4 and EMX1 loci (Fig. 1c). Importantly, the indel frequency was higher for HMGN1-GBE and lower for HMGN1-A8e at the tested loci (Fig. S1B). To further query the potentially adverse effects of HMGN1 fusion, the protein expression of base editors, gene expression of targeted loci, and cell viability were evaluated. The results indicated that the overexpression of HMGN1 did not significantly influence the expression of the base editors, gene expression of the targeted loci, or cell viability (Fig. S1C–E).

To further analyze the role of HMGN1 in optimizing GBE and A8e, we tested the HMGN1-GBE and HMGN1-A8e at more genomic loci in HEK293T cells. The data showed that HMGN1-GBE substantially increased the editing efficiency at eight genomic loci, especially at position C6 of the protospacer (Fig. 1d). Surprisingly, the increased indel frequency was not observed at all testing loci, and even decreased at the TET2-site1 (Fig. S1F). Our results also indicated that the HMGN1-A8e showed a modestly higher editing efficiency at most of the testing loci, especially at TET2-site4, with average increases of up to 37.40% (Fig. 1e). Furthermore, the indel frequency of HMGN1-A8e slightly decreased at several loci (Fig. S1G). More importantly, the HMGN1 fusions with GBE and A8e also enhanced the editing yield in HeLa cells (Fig. S1H) and exhibited substantial C-to-G and A-to-G transition in primary prostate carcinoma cells (Fig. S1I).

To further support the positive effects of HMGN1 and construct enhanced GBE and ABE variants, various components with diverse arrangements were tested with HMGN1. Firstly, the HMGN1 was fused to highly efficient miniCGBE variant[4], which incorporates the R33A mutation and deletes the UNG component of GBE. The data showed the HMGN1-miniCGBE enhanced the editing yield but also slightly increased the indel products at testing loci (Fig. S2A–C). We also attempted to replace the UNG component in the GBE to increase the editing yield. The Ung1 from *Saccharomyces cerevisiae*[10] and the Udgx from *Mycobacterium smegmatis*[29,30] were placed between APOBEC1 and Cas9 in the GBE system based on previously optimized arrangements[9] (Fig. S2D). The Udgx fusion showed a higher editing frequency at several genomic targets, especially in the PAM-proximal region (Fig. S2E). Finally, given that the pioneer factors were found to promote the chromatin accessibility and thereby enabled PAM-proximal base editing for CBE and GBE in previous research[27], we assumed that since HMGN1 also increases chromatin accessibility[31,32], it might similarly enable the PAM-proximal editing of A-to-G. Thus, we constructed an A8e variant where HMGN1 was placed between TadA and Cas9 (HMGN1-A8e-M; Fig. S2F) similar to the pioneer factors[27]. Our data showed that this reported A8e variant also exhibited a higher A-to-G yield of PAM-proximal adenines (Fig. S2G). Taken together, our data proved that HMGN1 efficiently enhanced the C-to-G conversion and modestly improved the efficiency of A-to-G editing.

### Construction of simultaneous C•G-to-G•C and A•T-to-G•C base editors

To extend the editing capacity for more applications of BEs, we speculate that the fusion of ABE and GBE could introduce a dual-functional base conversion. To test this, the GBE components including APOBEC1 and UNG were fused to A8e to construct a TABE-UNG (TadA8e-APOBEC1-Cas9-UNG; Fig. 2a). In addition, given that the UNG component might negatively affect the C-to-G transition[9], we also constructed a TABE system by omitting the UNG component (Fig. 2a). Then the TABE-UNG and TABE were tested at two genomic loci in

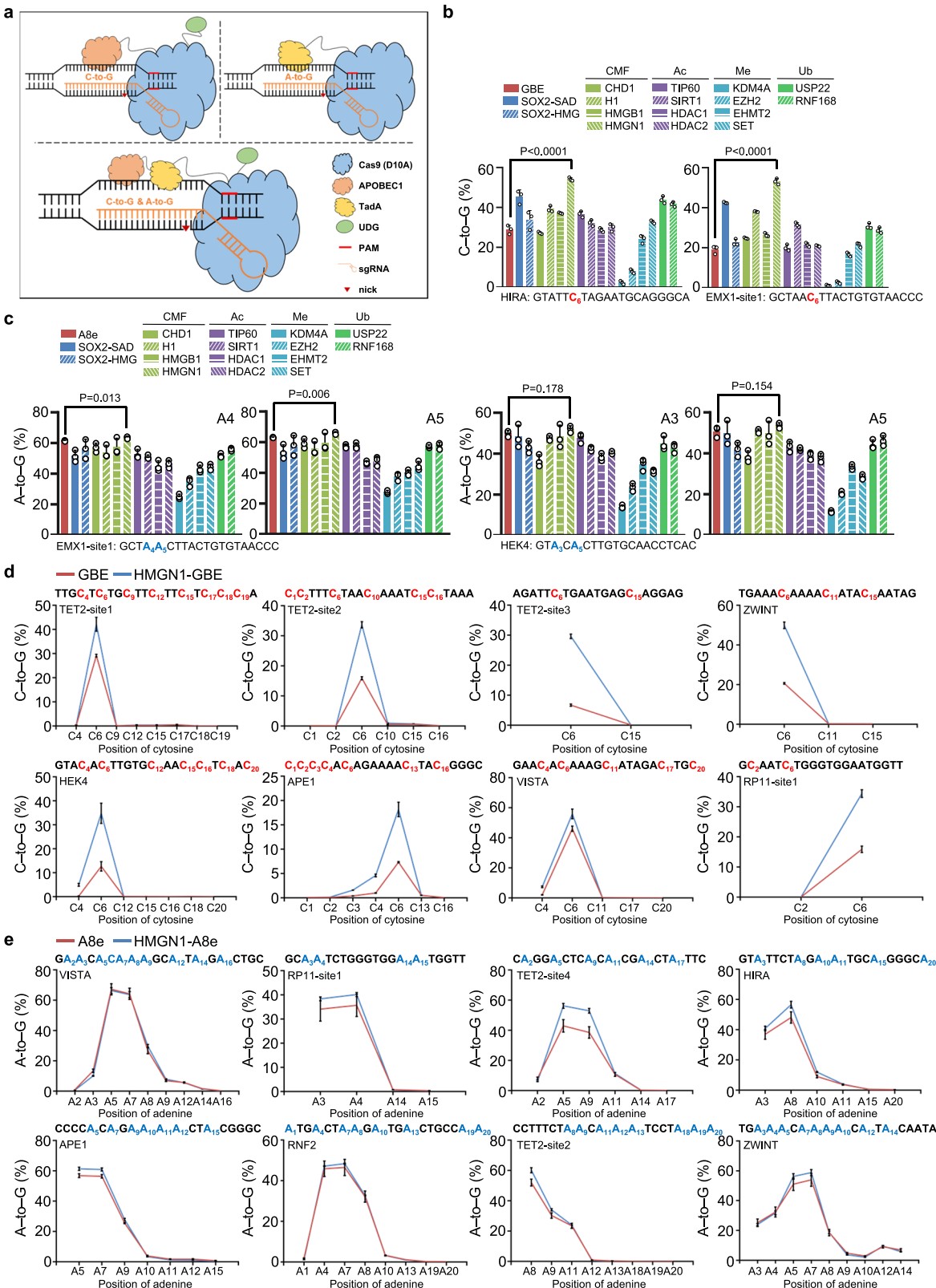

**Fig. 1 | Analysis of chromatin-associated factors for improving GBE and ABE editing. a** Schematic of GBE, ABE, and GGBE binding to DNA. **b** Base editing efficiency of C-to-G across GBE and GBE variants fused with 16 chromatin-associated factors at HIRA and EMX1-site1 loci in HEK293T cells. **c** Base editing efficiency of A-to-G across A8e and A8e variants fused with 16 chromatin-associated factors at HIRA and EMX1-site1 loci in HEK293T cells. **d** Base editing efficiency of C-to-G between GBE and HMGN1-GBE at eight genomic loci in HEK293T cells. **e** Base editing efficiency of A-to-G between A8e and HMGN1-A8e at eight genomic loci in HEK293T cells. CMF chromatin-associated factors, Ac acetylation, Me methylation, Ub ubiquitination. Mean ± SEM (**b**–**e**) of all individual values of sets of *n* = 3 independent replicates are shown. All statistical analysis for samples were conducted using unpaired Student's *t* test (two-tailed) in GraphPad Prism 8. Source data are provided as a Source Data file.

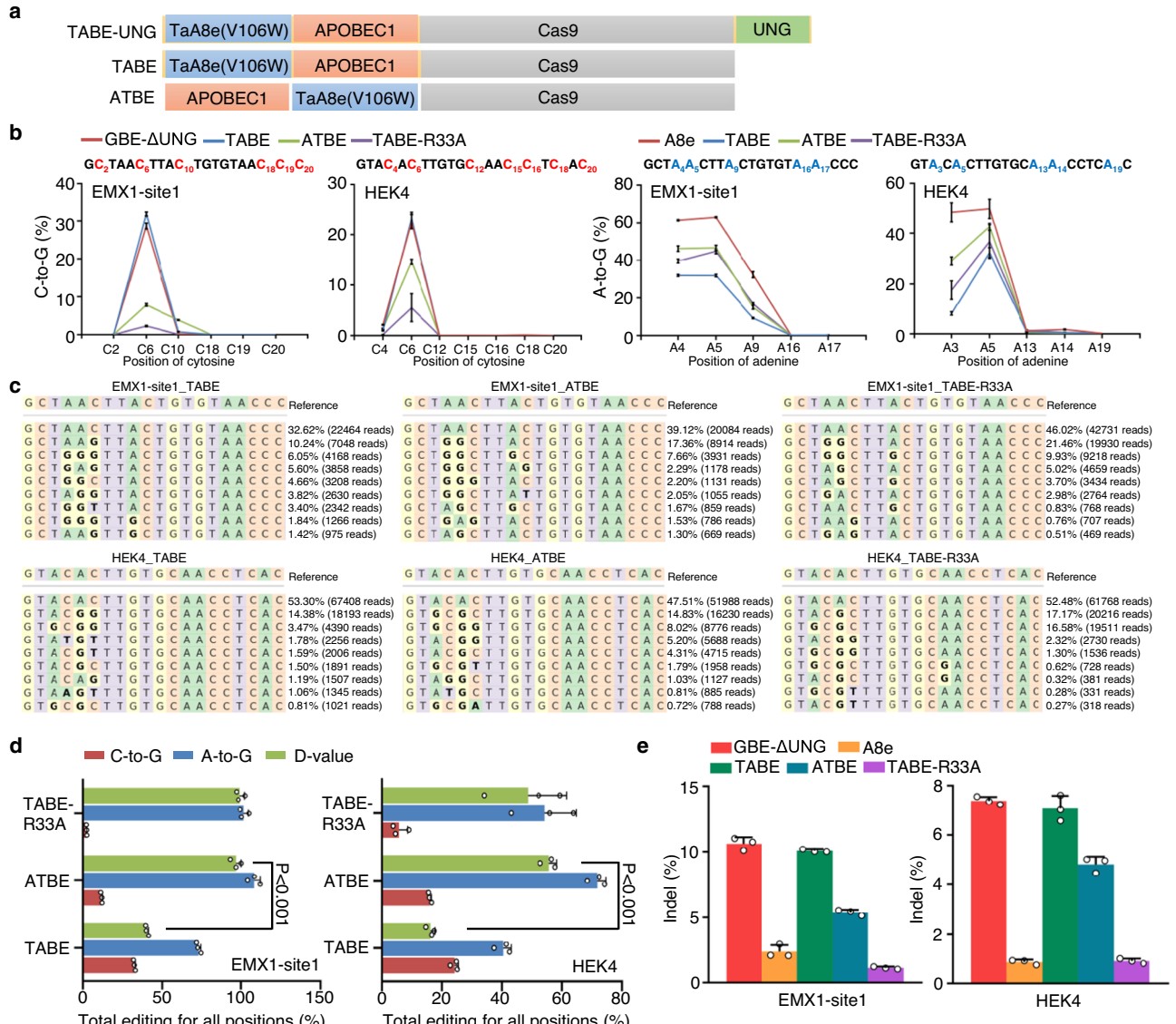

**Fig. 2 | Construction of simultaneous C•G-to-G•C and A•T-to-G•C base editors.** **a** Schematic of base editor variant with a fusion of APOBEC1, TadA8e, Cas9, and UNG. **b** Base editing efficiency of C-to-G (left) and A-to-G (right) across GBE-ΔUNG, A8e, TABE, ATBE and TABE (R33A) at EMX1-site1 and HEK4 in HEK293T cells. **c** The genotypes and reads and editing frequency of indicated genotypes across TABE, ATBE, and TABE (R33A) at EMX1-site1 (upper) and HEK4 (lower). **d** Comparison of editing frequency and D-value across TABE, ATBE, and TABE (R33A). Total editing is the sum of editing for all positions. **e** Comparison of indel frequency across GBE-ΔUNG, A8e, TABE, ATBE and TABE (R33A) at EMX1-site1 (left) and HEK4 (right) in HEK293T cells. Mean ± SEM (**b, d, e**) of all individual values of sets of $n = 3$ independent replicates are shown. Source data are provided as a Source Data file.

HEK293T cells. Although our data showed that TABE-UNG could simultaneously convert A-to-G and C-to-G, the editing efficiency was significantly lower than that of TABE (Fig. S3A).

To potentially improve the base editing efficiency, a ATBE (APOBEC1-TadA8e-Cas9) system was established (Fig. 2a), and the APOBEC1-R33A mutant[4,33] was also incorporated to improve the editing efficiency. To evaluate the balance of editing efficiency between A-to-G and C-to-G transition for dual functional base editors, we introduced a D-value, which is a measure of the difference in editing efficiency between A-to-G and C-to-G transition in our analysis. Despite ATBE induced a higher A-to-G conversion than TABE (Fig. 2b, c and S3B), the D-value was significantly increased, indicating an imbalanced A-to-G and C-to-G efficiency (Fig. 2d). In addition, the APOBEC1-R33A mutant with higher C-to-G editing was introduced in TABE, but we observed a relatively lower editing efficiency of C-to-G at two genomic loci for the TABE-R33A (Fig. 2b, c and S3B). Further, sequencing reads indicated that the TABE also exhibited the highest percentage of

concurrent C-to-G and A-to-G editing (Fig. S3C). Importantly, the indel frequency of TABE system retained a similar level to GBE (Fig. 2e). Thus, we chose the TABE system as the dual-function C-to-G and A-to-G base editor (GGBE1.0) for further investigation.

Next, we intended to optimize the GGBE by incorporating the chromosomal protein HMGN1 and other UNG proteins. HMGN1 and Udgx were integrated into the GGBE in different arrangements (GGBE1.1-1.5; Fig. 3a), and the resulting constructs were tested at two genomic loci in HEK293T cells. Our data indicated that the N-terminal fusion of HMGN1 (GGBE1.1) did not significantly improve the concurrent C-to-G and A-to-G base editing (Fig. 3b-d and S3D, E) or the D-value (Fig. 3e). With the addition of Udgx between APOBEC1 and Cas9 (GGBE1.5), both the D-value and base editing efficiency was decreased (Fig. 3b-e and S3D, E). Notably, we observed a higher PAM-proximal C-to-G editing with HMGN1 fusion between APOBEC1 and Cas9 (GGBE1.3), indicating an improvement of the editing window (Fig. 3b-d and S3D). Additionally, the indels of GGBE1.3 significantly

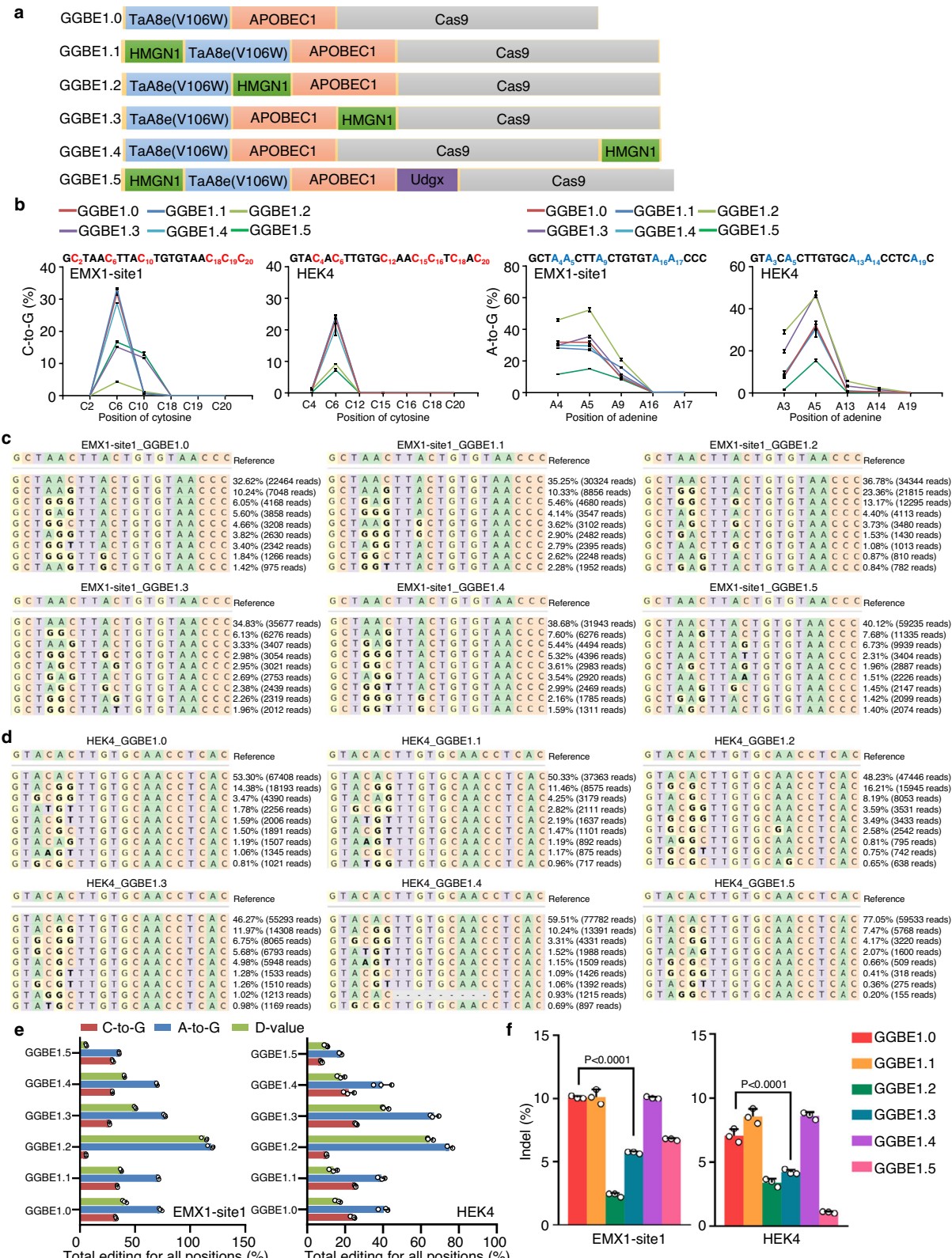

**Fig. 3 | Optimization of GGBE in diverse arrangements with HMGN1. a** Schematic of GGBE variants with fusion of HMGN1. **b** Base editing efficiency of C-to-G (left) and A-to-G (right) across GGBE1.0-GGBE1.5 at EMX1-site1 and HEK4 in HEK293T cells. **c** The genotypes and reads and editing frequency of indicated genotypes across GGBE1.0-GGBE1.5 at EMX1-site1. **d** The genotypes and reads and editing frequency of indicated genotypes across GGBE1.0-GGBE1.5 at HEK4. **e** Comparison of editing frequency and D-value across GGBE1.0-GGBE1.5. Total editing is the sum of editing for all positions. **f** Comparison of indel frequency across GGBE1.0-GGBE1.5 at EMX1-site1 (left) and HEK4 (right) in HEK293T cells. Mean ± SEM (**b**, **e**, **f**) of all individual values of sets of $n = 3$ independent replicates are shown. All statistical analysis for samples were conducted using unpaired Student's $t$ test (two-tailed) in GraphPad Prism 8. Source data are provided as a Source Data file.

decreased compared to GGBE1.0 (Fig. 3f). Considering that the GBE could also induce C-to-A/T conversion, the editing purity of cytosine was also evaluated. The data indicated that the proportion of C-to-G transition was lower in GGBE1.0 than in the original GBE at C6 of protospacer, whereas the GGBE1.3 induced a significantly higher C-to-G proportion for PAM-proximal cytosines at EMX1-site1 (Fig. S3F). Accordingly, GGBE1.0 and GGBE1.3 were selected for further verification of specific concurrent C-to-G and A-to-G base editing in subsequent analysis. Collectively, our data demonstrated that the GGBE could induce a concurrent C-to-G and A-to-G base editing, and the fusion of HMGN1 could expand the editing window of GGBE.

### Characterization and comparison of GGBE variants in mammalian cells

To further characterize the base editing features of GGBE and its variants, we then tested the GGBE1.0 and GGBE1.3 with another 12 genomic loci in HEK293T cells. We observed that A-to-G and C-to-G editing by GGBE1.0 and GGBE1.3 were mostly induced at positions 4–7 and 5–7 of the protospacer, respectively (Fig. 4a, b and S4A). The average editing efficiency by GGBE1.0 within the editing window were 16.52% for A-to-G conversion and 19.47% for C-to-G conversion, respectively. GGBE1.3 exhibited a moderately increased editing efficiency for both C-to-G and A-to-G transition at most tested genomic loci, especially for PAM-proximal base editing. The average editing efficiency of GGBE1.3 within the editing window were 25.94% and 21.82% for A-to-G and C-to-G conversion, respectively. For most tested gRNAs, the indel frequency of GGBE1.3 was modestly lower than that of GGBE1.0 (Fig. 4c). The percentage of concurrent C-to-G and A-to-G editing between GGBE1.0 and GGBE1.3 varied across these genomic loci, of which GGBE1.3 exhibited similar or higher simultaneous editing compared to GGBE1.0 (Fig. S4B). In addition, a higher proportion of C-to-G was also observed at several PAM-proximal cytosines in GGBE1.3, such as the PPP1R12C-site3-C9 and TET2-site2-C10 (Fig. S4C). Finally, the GGBE1.0 or GGBE1.3 were also tested in HeLa and primary prostate carcinoma cells with simultaneous C-to-G and A-to-G transition (Fig. S4D, E).

To further characterize the editing efficiency and specificity of GGBE, we compared it with the AGBE, a recently developed tool that was also reported to induce a concurrent C-to-G and A-to-G base conversion[17]. Considering the components and arrangements of dual base editors (Fig. 5a), the minAGBE-4 was selected for the comparison at three genomic loci in HEK293T cells. The data showed that the A-to-G and C-to-G transition in GGBE1.0 and GGBE1.3 was obviously higher than miniAGBE-4 at RP11-site1 and HEK4 (Fig. 5b, c and S5). A significantly higher proportion of C-to-G conversion was also observed in GGBE variants compared to miniAGBE-4 within the editing window (C5-C7), especially at HEK4-C6 site (Fig. 5d). More importantly, we found that the miniAGBE-4 could induce a significantly higher indel frequency than the GGBE variants (Fig. 5e). Overall, our analysis demonstrated that the GGBE1.0 and GGEB1.3 introduced a concurrent C-to-G and A-to-G base editing events with high editing efficiency and specificity.

### Off-target analysis of HMGN1-fused GBE, ABE, and GGBE in HEK293T cells

Given that the HMGN1 could alter chromatin accessibility, we further evaluated the off-target effects of HMGN1-fused GBE, ABE, and GGBE. To address the gRNA-dependent off-target effects, potential off-target (OT) sites were selected for analysis using Cas-OFFinder[34], or based on previously reported genomic loci[35], after which cumulative C-to-G or A-to-G editing was calculated. Our data showed a slightly higher off-target editing was observed in HMGN1-fused base editors (Fig. S6A, B).

Next, the effect of HMGN1-fused variants on Cas9-independent off-target DNA editing was characterized. A previously developed orthogonal R-loop assay[36] (Fig. S6C) was employed to evaluate off-target DNA editing at three genomic loci. Briefly, HEK293T cells were

co-transfected with plasmids encoding SpBE (*Streptococcus pyogenes* base editor) variants and an on-target sgRNA, along with a catalytically inactive SaCas9 (dSaCas9) and a SaCas9 sgRNA targeting a genomic locus unrelated to the SpBE on-target site. Then the Cas9-independent off-target editing was estimated based on detected editing efficiency in these dSaCas9-generated R-loops. Our data hinted that the HMGN1-fused base editors exhibited a higher editing frequency in one of the three R-loops (Fig. S6D). Finally, to measure the extent of cellular RNA editing by these base editors, HEK293T cells were transfected with the indicated base editors, after which the C-to-N and A-to-I mutation frequencies across the transcriptome were detected. The data showed that the HMGN1-fused variants did not induce a higher alteration of RNA editing (Fig. S6E), including the editing frequency (Fig. S6F) and a number of RNA single nucleotide variants (SNVs) (Fig. S6G). Collectively, our data indicated that the HMGN1-fused base editors might induce a modestly increased gRNA-dependent and Cas9-independent off-target DNA editing, but not RNA off-target editing.

### Potential application of GGBE in MNVs

Our results demonstrated that GGBE enables the efficient and concurrent conversion of C-to-G and A-to-G, thereby expanding the editing spectrum of conversion types. This expanded editing capability could be used to study and treat genetic diseases associated with MNVs, a newly identified category of genomic DNA sequence variants. Here, we identified a series of MNVs that could be rescued or created via GGBE through bioinformatics analysis (Supplementary Data 3). To further demonstrate the application value of GGBE, lentivirus-mediated knock-in[37] was employed to create model MNVs-associated loci on the chromosome of HEK293T cells, after which GGBE was introduced to evaluate the capacity to revert these model pathogenic MNVs. For example, Nakajo syndrome induced by the GG-to-CA mutation of PSMB8 (proteasome 20 S subunit beta 8)[38] was selected for investigation, which was characterized by skin eruption, splenomegaly, hyper γ-globulinemia, etc[39]. Our data indicated that the editing efficiency of targeted MNVs-associated clinical variants with GGBE1.0 ranged from 0.61% to 12.84% with simultaneous C-to-G and A-to-G editing (Fig. S7A, B). Overall, our analysis showed that the reported GGBE could be potentially employed to create cell models carrying clinically relevant MNVs, or even treat diseases by correcting these MNVs.

## Discussion

While the C-to-G base editors have been constructed, a dual functional C-to-G and A-to-G base editor has not been explored in detail. Further, the research focusing on how the chromatin microenvironment influenced genome editing is limited, especially for base editing. In this work, a chromatin-associated factor HMGN1 was found to improve both the base editing events of C-to-G and A-to-G. A dual deaminase-mediated base editor (GGBE) was constructed for simultaneous C-to-G and A-to-G conversion, which has the potential for studying and developing genetic therapies for MNVs.

Previous research demonstrated that several chromatin-modulating factors contributed to a similar improvement in Cas9 activity, including HMGN1, HMGB1, histone H1, and CHD1[20]. However, we found that these four factors had disparate editing enhancement effects on the C-to-G transition. The chromosomal protein HMGN1 exhibited the highest improvement for GBE, significantly higher than the previous GBE variant with pioneer factor SOX2, but the editing window did not alter. Considering that the C-to-G transition is formed via the DNA repair pattern of translesion DNA synthesis (TLS)[40], and HMGN1 was reported to enhance the rate of DNA repair via reducing the compaction of chromatin structure[31], the improvement of C-to-G was reasonable. However, HMGN1-fused A8e exhibited only a modest increase in A-to-G yield. Although HMGN1 might reduce the compacted chromatin to promote Cas9 binding, the editing effects of GBE

 

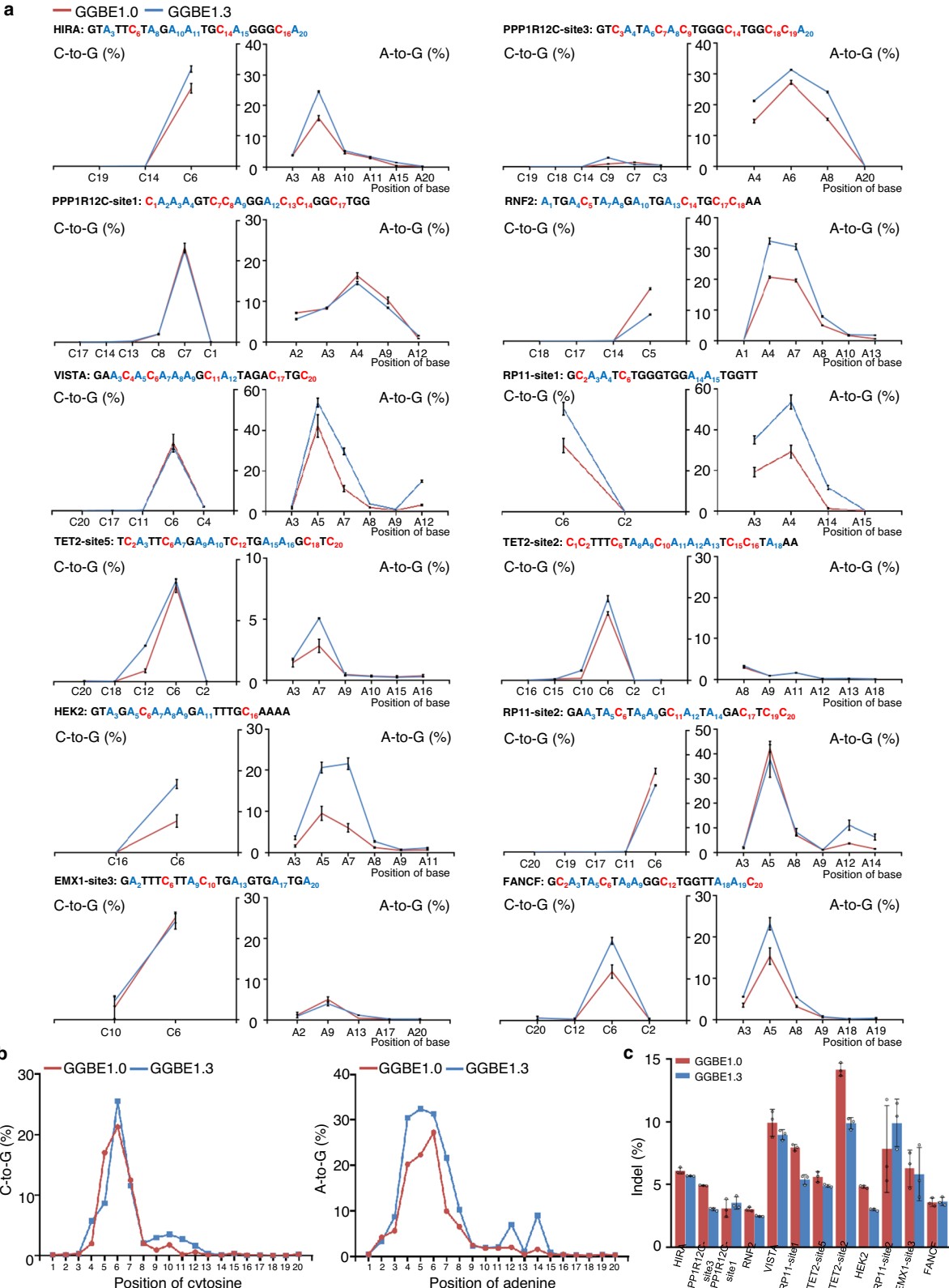

**Fig. 4 | Characterization of GGBE variants in mammalian cells. a** Comparison of editing efficiency between GGBE1.0 and GGBE1.3 at twelve endogenous genomic loci in HEK293T cells. **b** Average C-to-G and A-to-G base editing efficiency at C1-C20 (left) and A1-A20 (right) positions of protospacer from the twelve loci of GGBE1.0 and GGBE1.3. **c** Average indel frequency between GGBE1.0 and GGBE1.3 at 12 endogenous genomic loci in HEK293T cells. Mean ± SEM (**a, c**) of all individual values of sets of $n = 3$ independent replicates are shown. Source data are provided as a Source Data file.

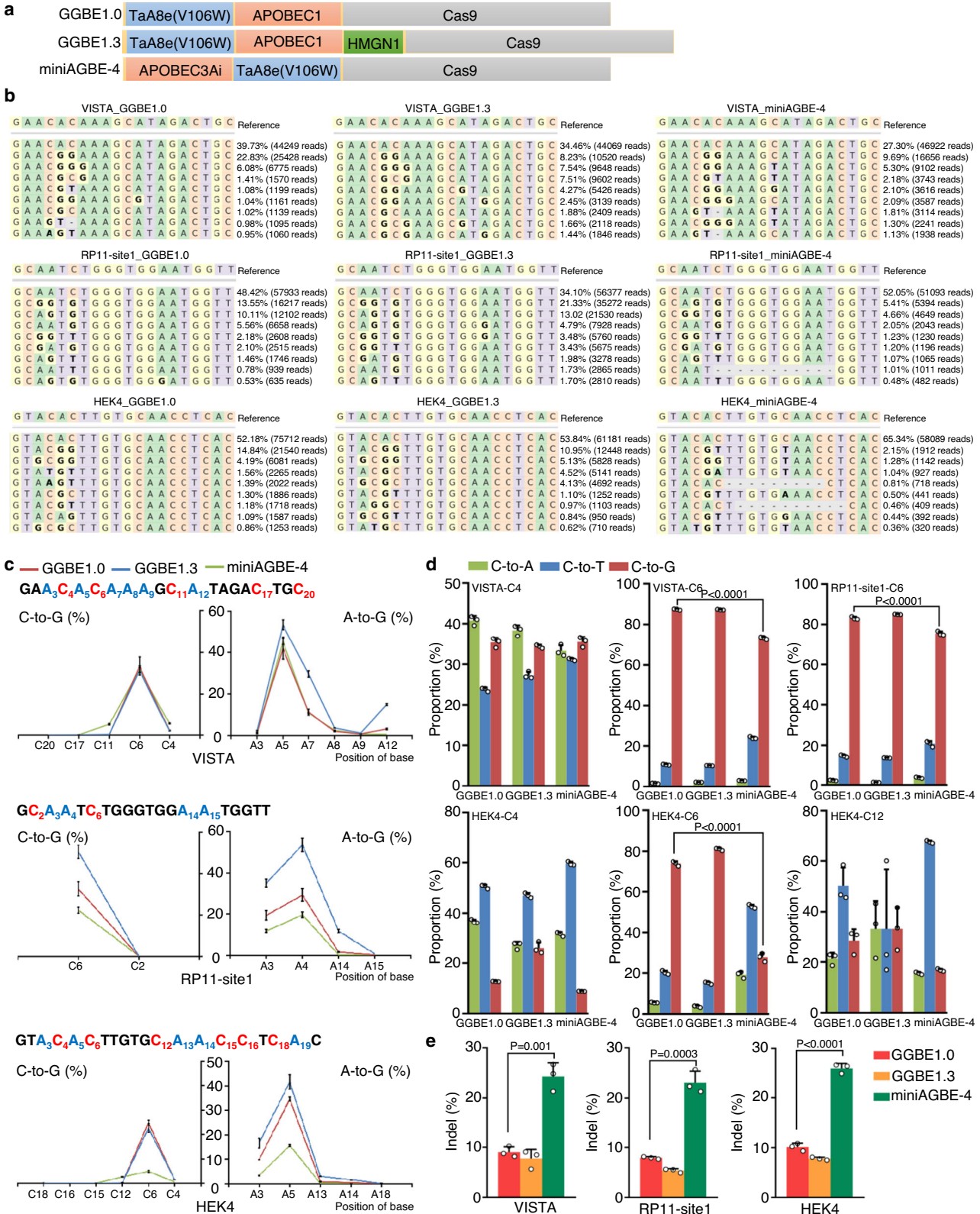

**Fig. 5 | Comparison between GGBE variants and AGBE in HEK293T cells.**
**a** Schematic of base editor variant of GGBE1.0, GGBE1.3, and mini-AGBE4. **b** The genotypes and reads and editing frequency of indicated genotypes across GGBE1.0, GGBE1.3, and mini-AGBE4 at VISTA, RP11-site1, and HEK4. **c** Comparison of editing efficiency across GGBE1.0, GGBE1.3 and mini-AGBE4 in HEK293T cells.
**d** Comparison of editing proportion of C-to-A/T/G across GGBE1.0, GGBE1.3 and

mini-AGBE4 in HEK293T cells. **e** Comparison of indel frequency across GGBE1.0, GGBE1.3 and mini-AGBE4 in HEK293T cells. Mean ± SEM (**c**–**e**) of all individual values of sets of $n = 3$ independent replicates are shown. All statistical analyses for samples were conducted using unpaired Student's $t$ test (two-tailed) in GraphPad Prism 8. Source data are provided as a Source Data file.

and ABE are also influenced by other molecular mechanisms. While C-to-G conversion is mediated via UDG and DNA polymerase of TLS in eukaryotes[40], A-to-G conversion is recognized as the inhibition of hypoxanthine excision[2]. Accordingly, while an open chromatin environment might facilitate the assembly of TLS-related repair factors to improve the C-to-G transition, the excision of hypoxanthine might not be influenced. However, we could not exclude other mechanisms accounting for the discrepancy between the effect of HMGN1 on the GBE and A8e.

A discrepant effect on the indel products between HMGN1-fused GBE and A8e was observed. The indel products of HMGN1-GBE increased at the majority of testing sites, while that of HMGN1-A8e did not. Notably, three TET2 sites tested by us in C-to-G conversion exhibited an obvious alteration of indel products. We assumed that the sequence context or the chromatin microenvironment might influence the indel products caused by a base editor. Additionally, histone-associated factors TIP60[41], USP22[42–44], and RNF168[45–47], which were observed to improve the C-to-G conversion, which was reported to be involved in DNA repair. Nevertheless, their function in base editing remains to be explored in more detail in the future. Notably, the histone methylase EZH2[48,49], which was reported to induce chromatin compaction, resulted in a significant decrease of C-to-G and A-to-G transition frequency, which further demonstrated the negative influence of a compacted chromatin microenvironment on base editing. More importantly, overexpression of HMGN1 did not induce any negative effects on the expression of the base editor, transcription of the targeted loci, or cell viability.

Next, to further confirm the effects of HMGN1 and construct improved base editor variants, diverse arrangements with HMGN1 and other efficient components were attempted. We found that the HMGN1-fused miniCGBE exhibited a higher C-to-G conversion efficiency, further indicating a positive role of HMGN1. In addition, given that the human UNG was reported to negatively affect the C-to-G transition, we also replaced it with Ung1 and Udgx to construct additional GBE variants. The HMGN1-GBE-Udgx exhibited a higher editing efficiency at the majority of testing loci, especially for the PAM-proximal cytosines. This finding was consistent with previous research[9]. Finally, HMGN1-A8e-M was observed to induce a higher base transition for PAM-proximal adenines, which was similar to the fusion of pioneer factors between Cas9 and deaminase. We hypothesized that the HMGN1 could improve the chromatin accessibility and also act as a long linker in this type fusion, hence increasing the binding capacity between deaminase and PAM-proximal bases. Overall, HMGN1 was found to positively influence the base editing efficiency of GBE and ABE.

Although an AGBE system has been constructed recently, its editing efficiency and specificity were relatively low[17]. To extend the genome editing possibility and improve the specificity, a GGBE catalyzing the concurrent C-to-G and A-to-G conversion was established by the fusion of GBE and A8e. The addition of UNG or Udgx to GGBE did not increase the editing yield, suggesting that the concurrent base editing might be negatively affected by the activity of glycosylase, which was consistent with the AGBE system[17]. To ensure the balance of two types of base conversion, we tested several arrangements of the deaminases, and introduced a more active APOBEC1 mutant into GGBE. The TadA8e-APOBEC1-Cas9 assembly was found to exhibit the optimal base editing pattern. To potentially improve the editing activity of GGBE, several variants incorporating HMGN1 were constructed. Unfortunately, the N-terminal fusion (GGBE1.1) was not found to improve the efficiency of concurrent base editing. These data indicate that simultaneous A-to-G and C-to-G base editing might potentially depend on a different molecular mechanism than single-base transition. Accordingly, the addition of HMGN1 to N-terminal did not result in higher concurrent editing, despite we could not exclude the possibility that GGBE1.1 could improve the efficiency at other

genomic loci with low intrinsic chromatin accessibility. Subsequently, a GGBE variant in which HMGN1 was placed between APOBEC1 and Cas9, called GGBE1.3, exhibited an improved editing window and decreased indel frequency. This fusion pattern increased the editing window possibly due to the role of a long linker and improved chromatin accessibility. We hypothesized that the decreased indel byproducts might be caused by the alteration of the editing window and also the chromatin accessibility. In addition, a comparison between GGBE and AGBE revealed that the former achieved more specific and efficient concurrent editing of C-to-G and A-to-G. Moreover, the indel frequency of AGBE was significantly higher than that of GGBE, which further demonstrated the superiority of our construct. More importantly, the HMGN1-fused base editor variants also induced slightly higher gRNA-dependent and Cas9-independent DNA off-target effects but not the RNA off-target effects. This phenomenon is in accord with the positive function of HMGN1 in base editing possibly due to the increased chromatin accessibility, which might not affect RNA editing caused by deaminases. Finally, since the GGBE could induce a dual-type base conversion, it could be applied for a spectrum of scenarios, such as the correction or creation of MNVs, and mutagenesis screens, as well as for programmable installation of transcription factor-binding sites. Thus, several clinically relevant MNVs were also constructed or corrected in cellular models with substantial editing efficiencies. However, the bystanders were also observed in these MNVs, which could be further optimized using deaminases with a narrow editing windows.

In summary, the construction of HMGN1-fused GBE, ABE, and GGBE has expanded the research frontiers of BEs, and enriched the base editing toolbox in mammalian cells.

## Methods

### Ethics statement
The study complied with all the ethical regulations for work with patients. The study was approved by the Ethics Committee of Union Hospital, Tongji Medical College, Huazhong University of Science and Technology and written informed consent was obtained from all patients.

### Cell culture, transfection, and CCK8 assay
Cell lines used in this research were purchased from the ATCC. HEK293T and HeLa cells were cultured in DMEM supplemented with 10% FBS in the humidified incubator equilibrated with 5% $CO_2$ at 37 °C. For transfected experiments, cells were seeded in 24-well plates (Corning, USA) and performed using polyethyienimine (Polysciences, USA) based on the manufacturer's instructions. Briefly, 600 ng of base editor and 300 ng of sgRNA-expressing plasmid were together transfected with 50 µl of Opti-MEM (Gibco, USA) containing 2.7 µl of polyethyienimine for 24 h. And then cells were replaced with fresh medium with 5 µg/ml puromycin (Merck, USA) for another 4 d. Finally, genomic DNA was extracted via QuickExtract DNA Extraction Solution (Epicentre, USA). On-target genomic regions (200 bp-300 bp) of interest were amplified by PCR for high-throughput DNA sequencing. The cell viability was evaluated via CCK8 analysis. Briefly, the HEK293T cells were seeded into 96-well plates and transfected with indicated base editors at approximately 60% confluency. Then after 48 and 72 h transfection, cell viability was measured by CCK8 reagent according to the manufacturer's instructions (DOJINDO, Japan).

### Culture, transfection, and assay of human primary prostate carcinoma cell
The primary cell was dissected and cultured as previously reported[50]. Briefly, fresh tissue biopsy samples were placed in media DMEM (Invitrogen, USA) with GlutaMAX (Invitrogen, USA), 100 U/ml penicillin, 100 µg/ml streptomycin (Gibcom USA), Primocin 100 g/ml (InvivoGen, USA), and 10 µmol/l ROCK inhibitor (Selleck Chemical Inc.,

USA) and washed. The dissected tissue was enzymatically digested with 250 U/ml of collagenase IV (Life Technologies, USA) and TrypLE express (Gibco) in a ratio 1:2 with Collagenase IV and then centrifuged. The pellet was washed and resuspended with prostate-specific culture media composed of DMEM/F12 (Invitrogen, USA) with GlutaMAX, 100 U/ml penicillin, 100 ₽g/ml streptomycin (Gibco, USA), Primocin 100 Mg/mL (InvitroGen, USA), B27 (Gibco, USA), N-Acetylcysteine 1.25 mM (Sigma-Aldrich, USA), Mouse Recombinant EGF 50 ng/ml (Invitrogen, USA), Human Recombinant FGF-10 20 ng/ml (Peprotech, USA), Recombinant Human FGF-basic 1 ng/ ml (Peprotech), A-83-01 500 nM (Tocris, USA), SB202190 10 kM (Sigma-Aldrich, USA), Nicoti-naminde 10 mM (Sigma-Aldrich, USA), (DiHydro) Testosterone 1 nM (Sigma-Aldrich, USA), PGE2 1 MM (R&D Systems, USA), Noggin conditioned media (5%) and R-spondin conditioned media (5%). The final resuspended pellet was combined with growth factor-reduced Matrigel (Corning, USA) in a 1:2 volume ratio and then pipetted onto a 24-well cell suspension culture plate.

The primary cells were transfected with indicated base editor and sgRNA plasmids using lipo3000 (GlpBio, USA). After 24 h transfection, cells were replaced with fresh medium with 1 µg/ml puromycin (Merck, USA) for another 5 d. Finally, genomic DNA was extracted via Quick-Extract DNA Extraction Solution (Epicentre, USA) and on-target genomic regions were amplified by PCR and analyzed with high-throughput DNA sequencing.

## Plasmid construction
Four chromatin-modulating proteins HMGB1, HMGN1, CHD1, and Histone H1 were synthesized by AZENTA. The other chromatin-associated factors were amplified with Phusion DNA polymerase (NEB, USA) from HEK293T cDNA library. PCR products were gel purified, digested with DpnI restriction enzyme (NEB, USA), and assembled via Gibson assembly based on the manufacturer's instructions. All gRNA-expression plasmids were assembled via Golden Gate with the protospacer sequence embedded in the primers, and *RNF2* sgRNA expression plasmids were used as the template[1]. The main primers are listed in Supplementary Data 1.

## Strains and culture conditions
*E. coli* Trans5α was used as the cloning host and cultured at 37 °C in lysogeny broth (LB, 1% (w/v) tryptone, 0.5% (w/v) yeast extract, and 1% (w/v) NaCl). 100 mg/L Ampicillin (Sigma, USA) were used for screen of positive cloning.

## Western blotting
The western blotting assay was performed as previously reported[48]. Briefly, cellular extracts from HEK293T cells were prepared with lysis buffer (50 mM Tris-HCl, pH8.0, 150 mM NaCl, 0.5% NP-40) for 30 min at 4 °C and then denatured for 10 min at 95 °C. The cell lysates were resolved using 10% SDS-PAGE gels and transferred onto acetate cellulose membranes. For incubation, membranes were incubated with Cas9 (Beyotime Biotechnology, dilution: 1:2000, Cat:AF0123, China) or GAPDH (ABclonal, dilution: 1:10,000, Cat:AC002, USA) antibodies at 4 °C overnight followed by incubation with a secondary antibody HRP-conjugated Affinipure Goat Anti-Mouse IgG (H + L) (Proteintech, dilution: 1:5000, Cat:AS003, USA). Immunoreactive bands were visualized using western blotting luminal reagent (Millipore, USA) according to the manufacturer's recommendation.

## High-throughput DNA sequencing of genomic DNA samples and data analysis
Next-generation sequencing library preparations and analysis were performed as previously reported[5]. Briefly, purified PCR fragments were treated in one reaction with End Prep Enzyme Mix for end repair, 5′ phosphorylation and dA tailing, which was followed by T-A ligation to add adapters to both ends, of which PCR products were purified and

quantified. Then the sequencing was carried out on Illumina HiSeq instrument according to the manufacturer's instructions.

Analysis of amplicon sequencing data were performed using CRISPResso2 v.2.0.45 in batch mode[28], with window parameters set to -wc 10 -w 20. The base conversion frequency was obtained with the output file 'Nucleotide_percentage_summary.txt' and indel rates was acquired with 'CRISPRessoBatch_quantification_of_editing_frequency.txt' for base editor experiments. All genomic loci and deep sequencing oligos of sgRNA are listed in Supplementary Data 2.

## RT PCR and Real-time RT PCR (qPCR)
Total cellular RNAs were isolated with SimplyP kit (Biospin, China) and used for the first strand cDNA synthesis with the Reverse Transcription System (TaKaRa, Japan). Quantitation of all gene transcripts was done by qPCR using Power SYBR Green PCR Master Mix and a Roche Roche LC96 sequence detection system with the expression of GAPDH as an internal control. The primer pairs used were: EMX1, 5′-TGTGCAT GTGCCTGGCTG-3′ (forward) and 5′-CTTGGCCACCAAGGACTCTA-3′ (reverse); HIRA, 5′-CTGGACTCTGAATGGGCTGG-3′ (forward) and 5′ GGCTAGGCTCTTGCCATAGG-3′ (reverse); and GAPDH, 5′-CTGGGC TACACTGAGCACC-3′ (forward) and 5′-AAGTGGTCGTTGAGGGCAA TG-3′ (reverse).

## RNA-seq analysis and SNVs calling
For transcriptome analysis, ~10[6] cells of each sample were collected and used for RNA extraction. The RNA-seq libraries were constructed and the high-throughput transcriptome sequencing was carried out with mina HiSeq instrument by the AZENTA company. For data analysis[33], qualified reads were mapped to the reference genome (Ensemble GRCh38) using STAR in 2-pass mode with default parameters. Then the Picard tool was used to sort and mark duplicates of the mapped BAM files, which were subject to split reads that spanned splice junctions, base recalibration, and variant calling with SplitNCigarReads, BaseRecalibrator, and HaplotypeCaller tools from GATK respectively.

The calling variants were filtered with default parameters using VariantFiltration tool from GATK. Variant loci in base editor over-expression experiments were filtered to exclude sites without high-confidence reference genotype calls in the control experiment. Base edits labeled as C-to-N comprise C-to-U/A/G edits called on the positive strand as well as G-to-A/U/C edits sourced from the negative strand. Base edits labeled as A-to-I comprise A-to-I edits called on the positive strand as well as T-to-C edits sourced from the negative strand.

## Analysis of potential targets for the correction or creation of MNVs by GGBE
A list of MNVs was obtained from previous reports[11,12], which were screened to detect disease-correcting or disease-creating modifications enabled by GGBE. Disease-correcting or -creating conversions are defined as having targetable C and A bases with matching T and G bases in the restricted position (base position 5–7 of the protospacer). Patterns for selected disease-correcting MNV codons include GNG > ANC, GGN > ACN, NGG > NAC, GNG > CNA, GGN > CAN, and NGG > NCA; whereas patterns for disease-creating include ACN > GGN, ANC > GNG, CAN > GGN, CNA > GNG, NAC > NGG and NCA > NGG. The filtered MNVs are listed in Supplementary Data 3.

## Lentivirus infection and base editing of MNVs-relevant targets
The generation of lentiviruses was conducted according to the previous reports[51,52]. Briefly, synthetic target sequences containing pathogenic point mutations together with psPAX2 and pMD2.G, were co-transfected into the packaging cell line HEK293T at a weight ratio of 3:2:1. Viral supernatants were collected 48 h later, clarified by filtration, and concentrated by ultracentrifugation. Then the concentrated viruses were used to infect 5–10[5] cells (20–30% confluence) in a 60-mm

dish with 5 mg/mL polybrene. Infected cells were selected by 4 µg ml⁻¹ blasticidin (Solarbio) to the culture medium. The target sequence-transduced HEK293T cells were then transfected with a mixture of plasmid encoding GGBE1.0 and targeted gRNA. After 5 days treatment with puromycin, cells were collected and the genomic DNA was subjected to deep sequencing to measure the editing efficiency of base editing.

## Statistics and reproducibility

Unless otherwise noted, all data are presented as means ± s.d. and analyzed with statistical methods from three independent experiments. The significance of the difference between the control and experiment group was calculated via student's $t$ test using GraphPad Prism 8. $P < 0.05$ was considered to be statistically significant.

## Reporting summary

Further information on research design is available in the Nature Portfolio Reporting Summary linked to this article.

## Data availability

HTS data generated in this study have been deposited in the NCBI Sequence Read Archive database under accession code PRJNA946328. Source data are provided with this paper.

## Code availability

Source code for CRISPResso2, STAR, Picard and GATK are available on github (https://github.com/pinellolab/CRISPResso2; https://github.com/alexdobin/STAR; https://github.com/broadinstitute/picard; https://github.com/broadinstitute/gatk).

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

## Acknowledgements
This research was financially supported by the National Natural Science Foundation of China (82203238, 32225031, 32171449), Tianjin Synthetic Biotechnology Innovation Capacity Improvement Project (TSBICIP-CXRC-034, TSBICIP-KJGG-017).

## Author contributions
C.Y., J.L., Z.C., X.Z., and C.B. designed the research and wrote the manuscript. C.Y. performed the experiments studies, and the computational studies, and analyzed data. Z.M., K.W., X.D., and M.H. performed experiments. Q.L. and X.Z. performed the computational studies and analyzed data.

## Competing interests
The authors declare no conflicts of interest.
