## [Peer Review File · Nature Communications]

Reviewers' Comments:

Reviewer #1:

Remarks to the Author:

In this study, the authors reported that chromosomal protein HMGN1 was positively affecting the base editing activity of ABE and GBE. Then, they employed HMGN1 to develop a dual-function GGBE, which would be useful for studying the multi-nucleotide variants-associated genetic disease. Although the combination of two base editor is not a novel strategy, this work focused on how the HMGN1 influence the base editing events, which provides a new direction in optimizing the base editing. Nevertheless, some concerns need to be solved before it is considered for publication.

Major concerns/suggestions:

1. The manuscript entitled "HMGN1 enhances CRISPR-directed dual-function A-to-G and C-to-G base editing (GGBE)", but in the RESULTS section, a lot of words and figures were used to describe the positive effect of HMGN1 on GBE and A8e. Please organize contents of the manuscript more logically.
2. Are there any literatures to demonstrate how HMGN1 influences the base editing events? (Line 99)
3. GGBE could convert simultaneous A and C to G with substantial editing efficiency. More reliable data such as genotypes, reads and frequencies of on-target sites should be provided to validate this claim.
4. The figure legends should be more informative. For example, the information of MNVs chosen in this Fig.S4 should be described in detailed, including WT and mutant sequences. And there was a formatting error in "AllHighlyPenetrant". The genotypes of the three loci by deep sequencing and the frequencies of editing should be provided
5. GBE/CGBE and AGBE have been reported to induce C-to-G, C-to-T, and C-to-A changes (Zhao et al., 2021, Kurt et al., 2021, Liang et al., 2022). How did GGBE perform in this aspect?
6. Safety evaluation of HMGN1-BEs is necessary, such as DNA/RNA off-target analysis, as HMGN1 could increase the chromatin accessibility.
7. In RESULT 3, it was necessary to elaborate on the reason for "placing HMGN1 between Tada and Cas9" rather than other positions and display the schematic of HMGN1-A8e-M.
8. In the Line 181, it had been demonstrated that "additional fusion of Udgx could further improve the editing events", why UNG was used to construct GGBE instead of Udgx (Line 212)?

Minor concerns/suggestions:

1. In the Line 61, authors pointed out "MNVs exhibited huge impact on the functional interpretation of genome data" based on Ref.11 and Ref.12. According to the description of Ref.11, "in order to understand the impact of these variants in clinical applications, we also annotated MNVs in 6072 sequenced individuals from rare disease families, including 4275 case samples. This resulted in 16 gained nonsense mutations and 110 changed missense MNVs with high CADD scores and low frequencies in gnomAD". This means that MNVs do not occur very often.
2. In the Line 126, there were fifteen variants, but in Fig.1, there were 16 variants.
3. In the Line 153, the conclusions that "HMGN1-A8e exhibited a significantly decreased indel frequency ..." and "HMGN1 could contribute to the editing events of A-to-G" were not reliable, as there was not very meaningful to compare between two frequencies that were around 2% and less than twice as difference.
4. In the Line 156, "C to G and A to G" should be consistent with the previous content.
5. In the Line 164, "the increased indel frequency was not observed at all loci" while P value was *** in 5/8 loci, and "indel frequency was significantly decreased at the TET2-site1" while P value was *. Please use the word significantly carefully. Similar problems exist elsewhere in the manuscript.
6. In the Line 177, "apobec1" should be "APOBEC1".
7. In the Line 218, it seems that Supplementary Fig.4A was not a correct figure.
8. In the Line 283, is there a better word to replace "insertion"?
9. In Fig.1, the legend of EZH2-GBE was not the same color as the column.
10. In Fig.4, it was confusing to mark different content with the same color.
11. In Fig.5E, this figure did not seem necessary.
12. In Fig.S2, it would be better to add the indel frequencies.
13. There were too many figures in the main text. Appropriate adjustments should be made to merge the parts that showed repetitive content and transfer the non-major parts to supplementary figures. For example, the results in Fig.1B and Fig.1C, Fig.1E and Fig.1F, Fig.S2A and Fig.S2B

showed the same conclusions, which were not necessary.

14. The data in the 392482_0_data_set_6961021_rjb898 was not intuitive. Please highlight the differences between the WT and mutant sequences. Besides, where was the Supplementary Table 4?

Reviewer #2:

Remarks to the Author:

Summary: The authors report the activity of different Cas9-derived genome base editors fused with chromatin remodeling proteins, and saw marked improvement of editing with the addition of HMGN1. It appears that the most novel part of the work might be dual C-to-G and A-to-G base editing, but in the text's current form it is unclear why this dual activity is useful. The other protein engineering work appears to be highly derivative (chromatin remodelers fused to previously characterized base editors). The work seems to have the potential to at least generate some interesting biological insights into the activity of base editors within chromatin, but the results are described in a very preliminary manner. Therefore publication in any form would be premature at this time.

MAJOR CONCERNS

It is very difficult to understand the significance and usefulness of simultaneous C-to-G and A-to-G base editing in general, which seems to be highlighted specifically in the introduction. The introduction describes previous relevant work, but it is not clear why making this particular type of editing more efficient is a significant hurdle for research and/or medicine.

The selection of the various base editors for the authors' study is not supported by a clear rationale.

The figures show quantitative data, but don't appear to be organized in any way that reveals anything biologically interesting. For instance, it is difficult to relate chromatin protein function with the outcomes of the assays.

The "positions" being referred to in the dot/line graphs need to be shown in a DNA sequence. It is difficult to immediately understand what the "positions" are referring to. Seeing the specific basepair positions within the context of the sgRNA binding site and surrounding sequence might help to explain differences in editing efficiency.

The plots with multiple points on the x-axes are missing x-axis labels.

The English grammar needs to be corrected throughout the manuscript.

Reviewer #3:

Remarks to the Author:

In this study, Yang and colleagues developed novel base editors with a higher efficiency or characterized by a dual activity (A>G and C to G). This work is interesting but there are several points to be addressed.

Main points:

- The best performing base editors should be tested in primary cells using delivery methods alternatively to plasmid transfection to demonstrate the potential application of these enzymes
- The Authors claim superiority of their enzyme over the ABE enzyme (Liang, NAR, 2022): they should perform a side-by-side comparison to prove it.
- When comparing original and modified enzymes, the Authors should check their expression/stability, at least by Western Blot
- Figure 1D: is it known that GBE generate a high frequency of Indels? The Authors should comment on that

- Figure 1: adding HMGN1 domain increase efficiency of GBE but not ABE8e: is this because the starting point (base editing efficiency obtained with the basic enzyme) is lower for GBE compared to ABE8e?
- Figure 1: what is the consequence or recruiting all these factors (or at least HMGN1) to the target genes? Does the expression change (even transiently) ?
- Page 3 of the result section, line 167: the Authors should refer to the original study generating the mini CGBE and explain the rationale of using a "mini" enzyme.
- Figure 2D: why Ung1 and Udgx are placed in a different position compared to UNG? Can these constructs be compared as they have a different structure?
- Figure 4: why HMGN1 has no effect on dual base editors?
- Page 6 of the results: The D value is reduced for ATBE: is it because GBE activity is reduced?
- Figure 5: why the Indels are lower with GGBE1.3 enzyme? Can the Authors comment on that?
- Supplementary Figure 4A: the Authors should evaluate the number of vector copies that they obtain to have a better idea of the efficiency of the system. In addition, it is not clear which enzymes have been used in these experiments.
- Is there any toxic effect of HMGN1 overexpression?
- When missing, Stats should be calculated
-

Minor points:

- Page 5 of the results, lines 203 and 204: I would tone down the conclusions: the new ABE variant have modestly higher editing efficiencies and not at all the loci (see Figure 1 and Suppl Figure 1), and they do not always show lower Indels
- In general, when the editing window is changed using newly developed enzyme, the Authors should suggest some potential explanations
- Please highlight in bold the name on the figure in the text to facilitate reading of the manuscript
- Page 5 of the results, line 218: I believe that the Authors refer to Suppl Figure 3 and not 4.
- It seems that panel 4F is not mentioned in the text
- There is only 1 panel in Supplementary Figure 4, the Authors can remove the A

Reviewer #1 (Remarks to the Author):

In this study, the authors reported that chromosomal protein HMGN1 was positively affecting the base editing activity of ABE and GBE. Then, they employed HMGN1 to develop a dual-function GGBE, which would be useful for studying the multi-nucleotide variants-associated genetic disease. Although the combination of two base editor is not a novel strategy, this work focused on how the HMGN1 influence the base editing events, which provides a new direction in optimizing the base editing. Nevertheless, some concerns need to be solved before it is considered for publication.

Major concerns/suggestions:

1. The manuscript entitled “HMGN1 enhances CRISPR-directed dual-function A-to-G and C-to-G base editing (GGBE)”, but in the RESULTS section, a lot of words and figures were used to describe the positive effect of HMGN1 on GBE and A8e. Please organize contents of the manuscript more logically.

Authors: Thanks for the reviewer's suggestion. We have included all parts about the positive effect of HMGN1 on GBE and A8e in RESULT 1 and FIGURE 1 to facilitate reading. The remaining FIGURE 2 to 5 showed the construction, optimization, characterization and comparison of GGBE, respectively. The remaining RESULT 2 to 5 showed the construction,

characterization, off-target effects and potential application of GGBE, respectively. We sincerely appreciate the reviewer's suggestion for making a more logical manuscript and clearer presentation of figures.

2. Are there any literatures to demonstrate how HMGN1 influences the base editing events? (Line 99)

Answer: Thanks for the reviewer's question. To the best of our knowledge, there is no work reporting HMGN1 fused base editors at time of our submission and this resubmission. We sincerely hope our response could solve your doubts.

3. GGBE could convert simultaneous A and C to G with substantial editing efficiency. More reliable data such as genotypes, reads and frequencies of on-target sites should be provided to validate this claim.

Answer: Thanks for the reviewer's suggestion. The genotypes, reads and frequencies have been provided in the revision (**Figure 2C, 3C, 3D, 5B, and S4A**). We sincerely appreciate the reviewer's suggestion for making a clearer presentation of figures.

For example, **Figure 2C**

4. The figure legends should be more informative. For example, the information of MNVs chosen in this Fig.S4 should be described in detailed, including WT and mutant sequences. And there was a formatting error in “AllHighlyPenetrant”. The genotypes of the three loci by deep sequencing and the frequencies of editing should be provided

Answer: Thanks for the reviewer’s suggestion. The information of MNVs was provided, including WT sequence, mutant sequence, genotypes, reads and frequency (Figure S7A). And the formatting error in “AllHighlyPenetrant” was corrected to “All Highly Penetrant” in Figure S7A. We sincerely appreciate the reviewer’s suggestion for making a clearer presentation of figures.

5. GBE/CGBE and AGBE have been reported to induce C-to-G, C-to-T, and C-to-A changes (Zhao et al., 2021, Kurt et al., 2021, Liang et al., 2022).

How did GGBE perform in this aspect?

Answer: Thanks for the reviewer's suggestion. The proportion of C-to-G, C-to-T and C-to-A of GGBE variants was added in the revision (**Figure S3D and S4B**). The data indicated that the proportion of C-to-G transition were lower in GGBE1.0 compared to original GBE at C6 of protospacer, whereas the GGBE1.3 induced a significantly higher C-to-G proportion for PAM-proximal cytosines (**Figure S3D**). Further, to demonstrate the superiority compared to the dual base editor-AGBE, the comparison including the C-to-G, C-to-T, and C-to-A changes between GGBE and AGBE was added in the revision. An obviously higher proportion of C-to-G conversion was also observed in GGBE variants compared to miniAGBE-4 within the editing window (C5-C7), especially at HEK4-C6 site (**Figure 5D**). We sincerely appreciate the reviewer's suggestion for making a more logical manuscript.

Line 239-244: "Considering that the GBE could also induce C-to-A/T conversion, the editing purity of cytosine were also evaluated. The data indicated that the proportion of C-to-G transition was lower in GGBE1.0 than in original GBE at C6 of protospacer, whereas the GGBE1.3 induced a significantly higher C-to-G proportion for PAM-proximal cytosines at EMX1-site1 (**Figure S3D**)."

Line 279-282: "A significantly higher proportion of C-to-G conversion was also observed in GGBE variants compared to miniAGBE-4 within the

editing window (C5-C7), especially at HEK4-C6 site (**Figure 5D**).”

For example, Figure S3D

6. Safety evaluation of HMGN1-BEs is necessary, such as DNA/RNA off-target analysis, as HMGN1 could increase the chromatin accessibility.

Answer: Thanks for the reviewer’s suggestion. The gRNA-dependent, Cas9-independent DNA and RNA off-target analysis was added in the **RESULT 4** of the revision. We sincerely appreciate the reviewer’s suggestion for making a more logical and substantial manuscript.

Line 289-319: “**4. Off-target analysis of HMGN1 fused GBE, ABE and GGBE in HEK293T cells**

Given that the HMGN1 could alter chromatin accessibility, we further evaluated the off-target effects of HMGN1 fused GBE, ABE and GGBE. To address the gRNA-dependent off-target effects, potential off-target (OT) sites were selected for analysis using Cas-OFFinder, or based on previously reported genomic loci, after which cumulative C-to-G or A-to-G editing were calculated. Our data showed a slightly higher off-target editing was observed in HMGN1 fused base editors (**Figure S6A and B**). Next, the effect of HMGN1 fused variants on Cas9-independent off-target

DNA editing was characterized. A previously developed orthogonal R-loop assay (**Figure S6C**) was employed to evaluate off-target DNA editing at three genomic loci. Briefly, HEK293T cells were co-transfected with plasmids encoding SpBE (Streptococcus pyogenes base editor) variants and an on-target sgRNA, along with a catalytically inactive SaCas9 (dSaCas9) and a SaCas9 sgRNA targeting a genomic locus unrelated to the SpBE on-target site. Then the Cas9-independent off-target editing was estimated based on detected editing efficiency in these dSaCas9-generated R-loops. Our data hinted that the HMGN1-fused base editors exhibited a higher editing frequency in one of the three R-loops (**Figure S6D**). Finally, to measure the extent of cellular RNA editing by these base editors, HEK293T cells were transfected with the indicated base editors, after which the C-to-N and A-to-I mutation frequency across the transcriptome were detected. The data showed that the HMGN1 fused variants did not induce a higher alteration of RNA editing (**Figure S6E**), including the editing frequency (**Figure S6F**) and number of RNA single nucleotide variants (SNVs) (**Figure S6G**). Collectively, our data indicated that the HMGN1 fused base editors might induce a modestly increased gRNA-dependent and Cas9-independent off-target DNA editing, but not RNA off-target editing.”

For example, **Figure S6C-D**

7. In RESULT 3, it was necessary to elaborate on the reason for “placing HMGN1 between TadA and Cas9” rather than other positions and display the schematic of HMGN1-A8e-M.

Answer: Thank you for the reviewer’s question. The schematic of HMGN1-A8e-M was added in the **Figure S2D** of the revision. And this type of fusion was based on previous positions of pioneer factors, of which the manuscript demonstrated that when pioneer factor was placed between APOBEC1 and Cas9, the editing window enlarged. And the detailed statement was added in the revision. We sincerely appreciate the reviewer’s suggestion for making a more logical manuscript.

Line 185-191: “Finally, given that the pioneer factors were found to promote the chromatin accessibility and thereby enabled PAM-proximal base editing for CBE and GBE in previous research, we assumed that since HMGN1 also increased chromatin accessibility, it might similarly enable the PAM-proximal editing of A-to-G. Thus, we constructed an A8e variant where HMGN1 was placed between TadA and Cas9 (HMGN1-A8e-M) similar to the pioneer factors (**Figure S2F**).”

Figure S2F

8. In the Line 181, it had been demonstrated that “additional fusion of Udgx could further improve the editing events”, why UNG was used to construct GGBE instead of Udgx (Line 212)?

Answer: Thank you for the reviewer’s question. This fusion with both the TadA8e and APOBEC1 was recognized as a novel base editor variant and we could not determine the role of UNG and Udgx in this different base editor compared to GBE. Thus, we started the construction of GGBE from the original GBE version and further tested the Udgx for the GGBE in our latter version of GGBE (GGBE1.5). We sincerely hope our response could solve your doubts.

Minor concerns/suggestions:

1. In the Line 61, authors pointed out “MNVs exhibited huge impact on the functional interpretation of genome data” based on Ref.11 and Ref.12. According to the description of Ref.11, “in order to understand the impact of these variants in clinical applications, we also annotated MNVs in 6072 sequenced individuals from rare disease families, including 4275 case samples. This resulted in 16 gained nonsense mutations and 110 changed missense MNVs with high CADD scores and low frequencies in gnomAD”.

This means that MNVs do not occur very often.

Answer: Thank you for the reviewer's suggestion. The statement was inappropriate in the first submitted manuscript. We have corrected the statement in the revision.

Line 68-70: "Significantly, the MNVs were identified as a clinically and biologically important class of genetic variation, influencing functional interpretation of genomic data."

2. In the Line 126, there were fifteen variants, but in Fig.1, there were 16 variants.

Answer: Thank you for the reviewer's suggestion. The statement was corrected to sixteen.

Line 131 "Then we constructed a series of engineered GBE and A8e variants fused with sixteen chromatin-associated factors"

3. In the Line 153, the conclusions that "HMGN1-A8e exhibited a significantly decreased indel frequency ..." and "HMGN1 could contribute to the editing events of A-to-G" were not reliable, as there was not very meaningful to compare between two frequencies that were around 2% and less than twice as difference.

Answer: Thank you for the reviewer's suggestion. The statement was corrected in the revision. We have deleted the "significantly" in the revision.

Further, given that the results obtained from eight genomic loci by HMGN1-A8e and four genomic loci by HMGN1-A8e-M in HEK293T or HeLa cells, we concluded that HMGN1 modestly improved A-to-G editing events.

Line 147-151: “The HMGN1 fused A8e (HMGN1-A8e) also exhibited the highest editing efficiency, but was comparable the control at the HEK4 and EMX1 loci (**Figure 1B**). Importantly, the indel frequency was higher for HMGN1-GBE and lower for HMGN1-A8e at the tested loci (**Figure S1B**).”

Line 193-195: “Taken together, our data proved that HMGN1 efficiently enhanced the C-to-G conversion and modestly improved the efficiency of A-to-G editing.”

4. In the Line 156, “C to G and A to G” should be consistent with the previous content.

Answer: Thank you for the reviewer’s suggestion. The statement was corrected in the revision.

Line 193-195: “Taken together, our data proved that HMGN1 efficiently enhanced the C-to-G conversion and modestly improved the efficiency of A-to-G editing.”

5. In the Line 164, “the increased indel frequency was not observed at all loci” while P value was *** in 5/8 loci, and “indel frequency was

significantly decreased at the TET2-site1” while P value was *. Please use the word significantly carefully. Similar problems exist elsewhere in the manuscript.

Answer: Thank you for the reviewer’s suggestion. The statement was corrected in the revision. And we changed the “***” to “P<0.05” or “P>0.05” for convenience of reading.

Line 162-164: “Surprisingly, the increased indel frequency was not observed at all testing loci, and even decreased at the TET2-site1 (**Figure S1F**)”

6. In the Line 177, “apobec1” should be “APOBEC1”.

Answer: Thank you for the reviewer’s suggestion. The statement was corrected in the revision.

Line 180-183: “The Ung1 from *Saccharomyces cerevisiae* and the Udgx from *Mycobacterium smegmatis* were placed between APOBEC1 and Cas9 in the GBE system based on previously optimized arrangements (**Figure S2D**).”

7. In the Line 218, it seems that Supplementary Fig.4A was not a correct figure.

Answer: Thank you for the reviewer’s suggestion. The statement was corrected in the revision.

Line 207-209: “Although our data showed that TABE-UNG could simultaneously convert A-to-G and C-to-G, the editing efficiency was significantly lower than that of TABE (**Figure S3A**)”

8. In the Line 283, is there a better word to replace “insertion”?

Answer: Thank you for the reviewer’s suggestion. The statement was corrected in the revision. “insertion” was changed to “knock-in”

Line 328-331: “To further demonstrate the application value of GGBE, lentivirus-mediated **knock-in** was employed to create model MNVs-associated loci on the chromosome of HEK293T cells”

9. In Fig.1, the legend of EZH2-GBE was not the same color as the column.

Answer: Thank you for the reviewer’s suggestion. The legend was corrected in the FIGURE 1 of revision.

10. In Fig.4, it was confusing to mark different content with the same color.

Answer: Thank you for the reviewer’s suggestion. The **Figure 2E** and **3F** was corrected to different color in the revision.

11. In Fig.5E, this figure did not seem necessary.

Answer: Thank you for the reviewer’s suggestion. The figure was deleted in the revision.

12. In Fig.S2, it would be better to add the indel frequencies.

Answer: Thank you for the reviewer's suggestion. The indel frequencies was added to **Figure S2C** of the revision.

13. There were too many figures in the main text. Appropriate adjustments should be made to merge the parts that showed repetitive content and transfer the non-major parts to supplementary figures. For example, the results in Fig.1B and Fig.1C, Fig.1E and Fig.1F, Fig.S2A and Fig.S2B showed the same conclusions, which were not necessary.

Answer: Thank you for the reviewer's suggestion. We have transferred the non-major parts to supplementary figures.

14. The data in the 392482_0_data_set_6961021_rjb898 was not intuitive. Please highlight the differences between the WT and mutant sequences. Besides, where was the Supplementary Table 4?

Answer: Thank you for the reviewer's suggestion. We have highlighted the differences between the WT and mutant sequences of Supplementary Table 3 in the revision, in which the wild type base was labeled in red color and mutant base was labeled in blue color. Further, we have carefully checked all the supplementary tables submitted in the revision for the accuracy of our manuscript.

Reviewer #2 (Remarks to the Author):

Summary: The authors report the activity of different Cas9-derived genome base editors fused with chromatin remodeling proteins, and saw marked improvement of editing with the addition of HMGN1. It appears that the most novel part of the work might be dual C-to-G and A-to-G base editing, but in the text's current form it is unclear why this dual activity is useful. The other protein engineering work appears to be highly derivative (chromatin remodelers fused to previously characterized base editors). The work seems to have the potential to at least generate some interesting biological insights into the activity of base editors within chromatin, but the results are described in a very preliminary manner. Therefore publication in any form would be premature at this time.

MAJOR CONCERNS

1. It is very difficult to understand the significance and usefulness of simultaneous C-to-G and A-to-G base editing in general, which seems to be highlighted specifically in the introduction. The introduction describes previous relevant work, but it is not clear why making this particular type of editing more efficient is a significant hurdle for research and/or medicine.

Answer: Thank you for the reviewer's question. We have added more

content explaining the importance of dual base editor and also the significance of efficiency and specificity of base editor. We sincerely appreciate the reviewer's suggestion for making a more logical manuscript.

Line 64-84: **“One base editor normally catalyzes a single type of base conversion, and it is not possible to simultaneously implement multiple editors in most scenarios. However, studies reported the discovery of MNVs (multi-nucleotide variants), which were recognized as two or more nearby variants existing on the same haplotype in an individual. Significantly, the MNVs were identified as a clinically and biologically important class of genetic variation, influencing functional interpretation of genomic data.** To extend the editing possibilities and study MNVs-associated genetic diseases, researchers have developed dual-function base editors, which enable simultaneous C-to-T and A-to-G conversion in mammalian cells. In addition, an AGE system was constructed using human APOBEC3A and TadA, which could catalyze four types of base conversions. These reconstructed BEs broadened the capability of base editing for applications in genetic therapy or gene regulation. **However, a series of MNVs with simultaneous C-to-G and A-to-G mutations remain poorly understood, and related genetic correction studies are hindered by the lack of genome editing tools. Additionally, high efficiency and specificity of base editors are indispensable for the construction of disease models and development**

of genetic therapies for mutational disease. However, the recently developed A₈BE showed low efficiency and specificity in performing concurrent C-to-G and A-to-G conversions.”

2. The selection of the various base editors for the authors' study is not supported by a clear rationale.

Answer: Thank you for the reviewer's question. While the C-to-G base editors have been constructed, a dual functional C-to-G and A-to-G base editor has not been explored in detail. Thus, the base editor specifically catalyzing the C-to-G and A-to-G conversion was considered based on the latest research. Then the original pattern (A₈BE and A₈e) of BEs which could convert C-to-G and A-to-G were selected in our study at the initial stage. And given that the A₈e-V106W mutant was reported to be similar to A₈e in editing efficiency but with reduced off-target effects, then the A₈e-V106W mutant was further selected to construct a safer dual base editor. We sincerely appreciate the reviewer's suggestion for making a more logical and clearer manuscript.

Line 125-131: “To further improve the efficiency and expand the editing scope of base editors, we intended to optimize them by integrating chromatin-associated factors and further construct a specific dual-function C-to-G and A-to-G base editor (GGBE). Accordingly, the nascent GBE was selected for investigation of C-to-G editing, while the latest highly efficient

A8e (ABE8e-V106W) with reduced off-target editing was selected to explore A-to-G editing.”

3. The figures show quantitative data, but don't appear to be organized in any way that reveals anything biologically interesting. For instance, it is difficult to relate chromatin protein function with the outcomes of the assays.

Answer: Thank you for the reviewer's suggestion. In the revision, we have reorganized all figures and results for more logical reading. And the several less important figures and contents were transferred to the supplemental materials. Further, the crucial data supporting our conclusion in the paper was labeled with the P value in the figures for revealing the biological significance, such as **Figure 3F** and **5D**. As for the **Figure 1**, the protein with similar function is marked in a same color with different patterns for the convenience of reading to understand the possible link between protein function and editing efficiency (**Figure 1A, 1B**). We sincerely appreciate the reviewer's suggestion for making a more logical manuscript and presentation of figures.

For example, **Figure 1A**

Figures

Figure 1

4. The “positions” being referred to in the dot/line graphs need to be shown in a DNA sequence. It is difficult to immediately understand what the “positions” are referring to. Seeing the specific base pair positions within the context of the sgRNA binding site and surrounding sequence might help to explain differences in editing efficiency.

Answer: Thank you for the reviewer’s suggestion. In the revision, we have added the context of the sgRNA for all genomic loci and further marked the targeted base with colors. We sincerely appreciate the reviewer’s suggestion for making a clearer presentation of figures.

For example, **Figure 1C**

5. The plots with multiple points on the x-axes are missing x-axis labels.

Answer: Thank you for the reviewer's suggestion. The x-axis labels "Position of bases" were added and carefully checked in the revision. We sincerely appreciate the reviewer's suggestion for making a clearer presentation of figures.

6. The English grammar needs to be corrected throughout the manuscript.

Answer: Thank you for the reviewer's suggestion. The revision has been corrected and carefully checked with native speaker. We sincerely appreciate the reviewer's suggestion for making a clearer manuscript.

Reviewer #3 (Remarks to the Author):

In this study, Yang and colleagues developed novel base editors with a higher efficiency or characterized by a dual activity (A>G and C to G). This work is interesting but there are several points to be addressed.

Main points:

- The best performing base editors should be tested in primary cells using delivery methods alternatively to plasmid transfection to demonstrate the potential application of these enzymes

Answer: Thank you for the reviewer's suggestion. The HMGN1-GBE, HMGN1-A8e and GGBE1.3 were tested with primary prostate carcinoma cells in the revision. The data indicated that these base editors exhibited substantial editing efficiency. We sincerely appreciate the reviewer's suggestion for making a more substantial manuscript.

Line 168-171: "More importantly, the HMGN1 fusions with GBE and A8e also enhanced the editing yield in HeLa cells (**Figure S1H**) and exhibited substantial C-to-G and A-to-G transition in primary prostate carcinoma cells (**Figure S1I**)."

Line 268-271: "Finally, the GGBE1.0 or GGBE1.3 were also tested in HeLa and primary prostate carcinoma cells with simultaneous C-to-G and A-to-G transition (**Figure S4C and D**)."

For example, **Figure S11**

- The Authors claim superiority of their enzyme over the AGBE enzyme (Liang, NAR, 2022): they should perform a side-by-side comparison to prove it.

Answer: Thank you for the reviewer's suggestion. In the revision, we chose the miniAGBE4 with similar arrangements to GGBE for the comparison. The data indicated that GGBE exhibited a significantly higher editing efficiency of concurrent editing of C-to-G and A-to-G at three testing genomic loci. Further, the GGBE also showed a higher specificity and lower indel products compared to AGBE. We sincerely appreciate the reviewer's suggestion for making a more logical and substantial manuscript. (**Figure 5**)

Line 272-284 : "To further characterize the editing efficiency and specificity of GGBE, we compared it with the AGBE, a recently developed tool which was also reported to induce a concurrent C-to-G and A-to-G base conversion. Considering the components and arrangements of dual base editors (**Figure 5A**), the minAGBE-4 was selected for the comparison at three genomic loci in HEK293T cells. The data showed that the A-to-G

and C-to-G transition in GGBE1.0 and GGBE1.3 was obviously higher than miniAGBE-4 at RP11-site1 and HEK4 (**Figure 5B, 5C and S5**). A significantly higher proportion of C-to-G conversion was also observed in GGBE variants compared to miniAGBE-4 within the editing window (C5-C7), especially at HEK4-C6 site (**Figure 5D**). More importantly, we found that the miniAGBE-4 could induce a significantly higher indel frequency than the GGBE variants (**Figure 5E**).”

For example, **Figure 5E**

- When comparing original and modified enzymes, the Authors should check their expression/stability, at least by Western Blot

Answer: Thank you for the reviewer’s suggestion. In the revision, the protein expression between original base editor (GBE, A8e) and HMGN1 fused variants were detected by Western Blot using Cas9 antibody (**Figure S1D**). We sincerely appreciate the reviewer’s suggestion for making a more logical and substantial manuscript.

Figure S1D

- Figure 1D: is it know that GBE generate a high frequency of Indels? The Authors should comment on that

Answer: Thank you for the reviewer's question. In the previous study¹⁻², the results showed that GBE generated a higher frequency of Indels. We sincerely hope our response could solve your doubts.

1. Kurt IC, et al. CRISPR C-to-G base editors for inducing targeted DNA transversions in human cells. *Nature biotechnology* 39, 41-46 (2021).

2. Zhao D, et al. Glycosylase base editors enable C-to-A and C-to-G base changes. *Nature biotechnology* 39, 35-40 (2021).

- Figure 1: adding HMGN1 domain increase efficiency of GBE but not ABE8e: is this because the starting point (base editing efficiency obtained with the basic enzyme) is lower for GBE compared to ABE8e?

Answer: Thank you for the reviewer's question. We assumed that the lower starting point of GBE could partially accounted for the higher increased efficiency of GBE. And the differential working mechanism between GBE and A8e might also lead to the difference. We sincerely hope our response could solve your doubts.

Line 364-370: "While C-to-G conversion is mediated via UDG and DNA polymerase of TLS in eukaryotes, A-to-G conversion is recognized as the inhibition of hypoxanthine excision. Accordingly, while open chromatin environment might facilitate the assemble of TLS-related repair factors to

improve the C-to-G transition, the excision of hypoxanthine might not be influenced. However, we could not exclude other mechanisms accounting for the discrepancy between the effect of HMGN1 on the GBE and A8e.”

- Figure 1: what is the consequence or recruiting all these factors (or at least HMGN1) to the target genes? Does the expression change (even transiently) ?

Answer: Thank you for the reviewer’s question. The EMX1 and HIRA expression were detected with control and HMGN1 fusion in HEK293T cells via qRT-PCR with GBE/HMGN1-GBE and A8e/HMGN1-A8e (**Figure S1E**). And we did not observe any significant alteration of gene expression. We sincerely appreciate the reviewer’s suggestion for making a more logical and substantial manuscript. We sincerely hope our response could solve your doubts.

Figure S1E

- Page 3 of the result section, line 167: the Authors should refer to the original study generating the mini CGBE and explain the rational of using

a “mini” enzyme.

Answer: Thank you for the reviewer’s suggestion. The reference was added for the miniCGBE in the revision. Although the base editor was labeled with “mini”, the miniCGBE exhibited a higher editing efficiency and lower indel products. Hence, to further support the positive role of HMGN1 and construct the better GBE variants, we selected the miniCGBE variant to verify the HMGN1 and also construct a better base editor. We sincerely appreciate the reviewer’s suggestion for making a more logical manuscript. Line 174-176: “Firstly, the HMGN1 was fused to highly efficient miniCGBE variant⁴, which incorporates the R33A mutation and deletes the UNG component of GBE.”

- Figure 2D: why Ung1 and Udgx are placed in a different position compared to UNG? Can these constructs be compared as they have a different structure?

Answer: Thank you for reviewer’s question. The Ung1 and Udgx was also placed in the C-terminal in our work but the editing efficiency was similar and even lower than HMGN1-GBE (data not shown). Here, our goal was to construct one base editor with better editing function. Given that the previous work demonstrated the optimal position of glycosylase in C-to-G base editor, we then aimed to construct better C-to-G base editors with Ung1 and Udgx in an optimal position. The comparison was performed in

different version of C-to-G base editors, not the different function of glycosylase. We sincerely hope our response could solve your doubts.

- Figure 4: why HMGN1 has no effect on dual base editors?

Answer: Thank you for reviewer's question. Our data demonstrated that the HMGN1 fused GGBE at N-terminal (GGBE1.0) showed no effect on dual base editors at two genomic loci. But the GGBE1.3 (HMGN1 was placed between APOBEC1 and Cas9) exhibited a higher editing efficiency and wider editing window at most of the twelve genomic loci (**Figure 4A and 4B**). We assume that the uncertain functional mechanism of dual base editor might account for this phenomenon. However, the GGBE1.0 might also work at several genomic loci with tough chromatin accessibility. The detailed content explaining this difference was added in the revision. We sincerely hope our response could solve your doubts.

Line 416-429: "To potentially improve the editing activity of GGBE, several variants incorporating HMGN1 were constructed. Unfortunately, the N-terminal fusion (GGBE1.1) was not found to improve the efficiency of concurrent base editing. These data indicate that simultaneous A-to-G and C-to-G base editing might potentially depend on a differential molecular mechanism than single base transition. Accordingly, the addition of HMGN1 to N-terminal did not result in higher concurrent editing, despite we could not exclude the possibility that GGBE1.0 could improve

the efficiency at other genomic loci with low intrinsic chromatin accessibility. Subsequently, a GGBE variant in which HMGN1 was placed between APOBEC1 and Cas9, called GGBE1.3, exhibited an improved editing window and decreased indel frequency. This fusion pattern increased the editing window possibly due to the role of a long linker and improved chromatin accessibility.”

- Page 6 of the results: The D value is reduced for ATBE: is it because GBE activity is reduced?

Answer: Thank you for reviewer’s question. We have corrected the error “The D-value is reduced for ATBE” to “The D-value is increased for ATBE”. Our data showed that the GBE activity is reduced and A8e activity is increased, then the D value increased. We sincerely hope our response could solve your doubts.

Line 216-219: “Despite ATBE induced a higher A-to-G conversion than TABE (**Figure 2B, 2C and S3B**), the D-value was significantly increased, indicated an imbalanced A-to-G and C-to-G efficiency (**Figure 2D**).”

- Figure 5: why the Indels are lower with GGBE1.3 enzyme? Can the Authors comment on that?

Answer: Thank you for reviewer’s question. Actually, the decreased indels were not observed at all testing loci. We assumed that the decreased indel

byproducts might be induced due to the alteration of the editing window and also the chromatin accessibility. The alteration of editing window could change the editing efficiency of the protospacer and the altered chromatin microenvironment could change the DNA repair process, thus together affects the indel products. We sincerely hope our response could solve your doubts.

Line 429-431: “We hypothesized that the decreased indel byproducts might be caused by alteration of the editing window and also the chromatin accessibility.”

- Supplementary Figure 4A: the Authors should evaluate the number of vector copies that they obtain to have a better idea of the efficiency of the system. In addition, it is not clear which enzymes have been used in these experiments.

Answer: Thank you for reviewer’s question. The number of vector copies (VCN=2.32) was determined via qRT-PCR and added in the **FIGURE S7A** of the revision for better understanding the efficiency of system. Further, the genotype, reads and frequencies of indicated genotypes were provided in the revision to evaluate the editing efficiency of this system. And the detailed base editors (**GGBE1.0**) were added in the figure content (**Figure S7A**). We sincerely hope our response could solve your doubts.

Figure S7A

Figure S7

- Is there any toxic effect of HMGN1 overexpression?

- When missing, Stats should be calculated

Answer: Thank you for reviewer's question. The cell viability was tested between original base editors and fused variants. The data showed that the overexpression of HMGN1 did not show higher toxic effects (**Figure S1C**). We sincerely appreciate the reviewer's suggestion for making a more logical and substantial manuscript.

Figure S1C

Minor points:

- Page 5 of the results, lines 203 and 204: I would tone down the

conclusions: the new ABE variant have modestly higher editing efficiencies and not at all the loci (see Figure 1 and Suppl Figure 1), and they do not always show lower Indels

Answer: Thank you for reviewer's suggestion. We have corrected the conclusion according to the reviewer's suggestion.

Line 193-195: "Taken together, our data proved that HMGN1 efficiently enhanced the C-to-G conversion and modestly improved the efficiency of A-to-G editing."

- In general, when the editing window is changed using newly developed enzyme, the Authors should suggest some potential explanations

Answer: Thank you for reviewer's suggestion. The discussion was added in the revision.

Line 397-402: "Finally, HMGN1-A8e-M was observed to induce a higher base transition for PAM-proximal adenines, which was similar to the fusion of pioneer factors between Cas9 and deaminase. We hypothesized that the HMGN1 could improve the chromatin accessibility and also act as a long linker in this type fusion, hence to increase binding capacity between deaminase and PAM-proximal bases."

Line 429-431: "We hypothesized that the decreased indel byproducts might be caused by alteration of the editing window and also the chromatin accessibility."

- Please highlight in bold the name on the figure in the text to facilitate reading of the manuscript

Answer: Thank you for reviewer's suggestion. We have modified the name of figure in the revision.

- Page 5 of the results, line 218: I believe that the Authors refer to Suppl Figure 3 and not 4.

Answer: Thank you for reviewer's suggestion. We have corrected in the revision.

Line 207-209: "Although our data showed that TABE-UNG could simultaneously convert A-to-G and C-to-G, the editing efficiency was significantly lower than that of TABE (**Figure S3A**)"

- It seems that panel 4F is not mentioned in the text

Answer: Thank you for reviewer's suggestion. We have corrected in the revision. The panel 4F is changed to 3A.

Line 226-229: "Next, we intended to optimize the GGBE by incorporating the chromosomal protein HMGN1 and other UNG protein. HMGN1 and Udgx were integrated into the GGBE in different arrangements (GGBE1.1-1.5; **Figure 3A**)"

- There is only 1 panel in Supplementary Figure 4, the Authors can remove the A

Answer: Thank you for reviewer's suggestion. We have added a FIGURE S4B in the revision and carefully checked all figures.

Reviewers' Comments:

Reviewer #1:

Remarks to the Author:

The authors have made careful revisions in response to the comments and have addressed most of the concerns I raised properly. The quality of the revised manuscript has been significantly improved. However, there are still some minor details that need to be optimized in this version of the manuscript. I recommend publication after revision.

Minor concerns/suggestions:

1. In Lines 64-66, it is not an accurate statement, as many dual base editors enable both cytosine and adenine base conversions, while the single base editors also catalyze different types of base conversion in many cases (such as AYBE and GBE).

Ref. Kim HS, Jeong YK, Hur JK, Kim JS, Bae S. Adenine base editors catalyze cytosine conversions in human cells. *Nat Biotechnol.* 2019;37(10):1145-1148.

2. In the INTRODUCTION section, the literature of MNVs with C-to-G and A-to-G should be listed to show the significance and usefulness of GGBE rather than other types of dual base editors.

3. In Lines 106-107, it appears that the sentence uses an incorrect form of the verbs, corroborating and enhancing.

4. In Fig.S1B, some information is missing in the figure legends.

5. In Fig.S2C, the colors of the bar plots do not match the legends.

6. In Fig.2C, Fig.3C-D, and Fig.S4A, it is necessary to list the specific percentage of simultaneous A-to-G and C-to-G conversions.

Ref. Li C, Zhang R, Meng X, et al. Targeted, random mutagenesis of plant genes with dual cytosine and adenine base editors. *Nat Biotechnol.* 2020;38(7):875-882.

7. In Fig.2D and 3E, what does the meaning of "total editing (%)" as the frequency of A-to-G is more than 100% in many cases?

8. In Fig.3D, it appears that a type of genotype is missing in HEK4_GGBE1.5.

9. In Fig.S4C-D, the figure legends are wrong, and it cannot be found a comparison in Fig.S4D.

10. In Lines 327-328, a brief explanation is needed to introduce the clinical symptoms of the MNVs chosen in this study, which can highlight the value of GGBE.

Reviewer #2:

Remarks to the Author:

The authors were very responsive to my concerns. The scientific context added to the manuscript makes the significance much clearer. The addition of DNA target sequences make the context of the base edits more clear. I have the following requests for minor revisions:

The manuscript could be greatly improved by including a simple cartoon model of the GBE and A8e complexes bound to DNA (see AC Komor et al, *Nature* 2016, Figure 1 for an example, but the inclusion of the base conversion steps is not needed for the authors' submitted manuscript). This can be added as a new Figure 1, or as an additional panel in the current Figure 1 if there is space. Earlier base editors have used dCas9 or Cas9n (nickase). It is important to make it clear to the reader that Cas9 which induces DSB's is being used here, and also any relevant point mutations in other modules of the fusion protein. In the cartoon, the cut site should be shown so that it is clear where the target bases are relative to the DSB site. In general, it is better to present such models within a paper rather than expecting the reader to look to previous papers for such illustrations.

Figure 1. Please include a p-value to compare HMGN1-GBE to GBE and HMGN1-A8e to A8e. The differences for the latter pair do not seem very significant.

Figures (general) - Since the editors use a DSB-inducing Cas protein, the position of the DSB should be shown as a vertical line in all of the sequences.

Reviewer #3:

Remarks to the Author:

The Authors have addressed most of my comments, I still have a few remarks on Figure S7A: (1) VCN is the same for the 3 target regions, did the authors use a unique LV harboring the elements. Please specify it
(2) I do not understand the rational of testing GGBE1.0 which does not contain HMBG1, does HMBG1 increase this efficiency?
(3) Bystander edits are visible. The Authors should comment on the effect of these unwanted base conversions.